# Spatial Molecular AlO Temperature Distributions in Laser-Induced Plasma

**David M. Surmick** [1,*] , **Daryl J. Dagel** [2] **and Christian G. Parigger** [3]

[1] Physics and Applied Physics Department, University of Massachusetts of Lowell, 265 Riverside Street, Lowell, MA 01854, USA

[2] Sandia National Laboratories, 1515 Eubank, Albuquerque, NM 87115, USA; djdagel@sandia.gov

[3] Physics and Astronomy Department, University of Tennessee, University of Tennessee Space Institute, Center for Laser Applications, 411 B.H. Goethert Parkway, Tullahoma, TN 37388, USA; cparigge@tennessee.edu

\* Correspondence: david_surmick@uml.edu; Tel.: +1-978-934-3822

**Abstract:** Spatially resolved, line-of-sight measurements of aluminum monoxide emission spectra in laser ablation plasma are used with Abel inversion techniques to extract radial plasma temperatures. Contour mapping of the radially deconvolved signal intensity shows a ring of AlO formation near the plasma boundary with the ambient atmosphere. Simulations of the molecular spectra were coupled with the line profile fitting routines. Temperature results are presented with simultaneous inferences from lateral, asymmetric radial, and symmetric radial AlO spectral intensity profiles. This analysis indicates that shockwave phenomena in the radial profiles, including a temperature drop behind the blast wave created during plasma initiation were measured.

**Keywords:** molecular spectroscopy; diatomic spectroscopy; Abel transform; plasma spectroscopy; laser-induced breakdown spectroscopy; laser-induced plasma; plasma dynamics

## 1. Introduction

The act of tightly focusing laser radiation on to a target volume initiates plasma coupled with the propagation of a shockwave. Optical spectroscopy measurements of the dynamics are made in an effort to understand the chemistry kinetics of laser ablation plasma. Developments of line-of-sight diagnostics for such systems are advantageous due to the numerous applications that are of interest, including scaling laws between high explosive events and laser-induced plasma shock, temperature, and electron density phenomena [1,2], molecular formation of nanocluster formation in laser plasma plumes [3], pulsed laser vapor deposition [4,5], and analytical applications of laser-induced breakdown spectroscopy (LIBS) [6,7]. In each of these applications the distribution of atomic and molecular emissions and the associated plasma state quantities, i.e., electron density and temperature, are of interest. In particular, correlations between spatial distributions, molecular emission intensities and subsequent temperature inferences and the shock phenomena associated with laser-induced plasma are desirable. This information is most desirable for laser-produced plasma at atmospheric conditions, as opposed to vacuum conditions, where the plasma and shock dynamics are more complex due to plume splitting and confinement.

We consider asymmetric Abel inverted, diatomic molecular aluminum monoxide (AlO) emission spectra measured from laser-produced plasma on the surface of an Al target sample in ambient laboratory air as a method for extracting further spatial distribution information about laser ablation plasma. Abel transformations and the more generalized Radon transformation relate lateral, line-of-sight measurements of the plasma emission intensity to the radial intensity distribution through

well-known integral transformations [8,9]. This is done in an effort to extract information about the radial distributions of the molecular emission. Radon transformation techniques are commonly applied to laser ablation scenarios [10,11], however, due to the more general nature of the Radon transformation, measurements made along multiple lines-of-sight are typically required. The Abel inversion assumes some type of symmetry, which alleviates the need for multiple line-of-sight measurements. For a cylindrically symmetric emission source, the Abel transformation is given as

$$I(z,\lambda) = 2 \int_r^R I(r,\lambda) \frac{rdr}{\sqrt{r^2 - z^2}}. \tag{1}$$

Here, $I(z,\lambda)$ is the lateral spectral measurement and $I(r,\lambda)$ is the radial reconstruction. The upper limit of the integral, $R$, is often carried out to be much greater than the extent of the plasma. The integral pre-factor of 2 indicates the plasma symmetry along the radial and lateral axes.

The desire in attempting to apply Abel reconstructions over the Radon deconvolutional methods is to adopt a procedure that is more widely applicable given the ease with which a single lateral, spectral measurement can be made along a single axis of the plasma plume. Further, numeric methods for implementing Abel transformation are plenty and can be applied with little computational expense. Asymmetries are included by using a procedure developed by Blades [12], originally applied to inductively coupled plasma with slightly asymmetric spectral emissions. This method focusses on the creation of a symmetric profile from two plasma emission halves that are symmetric, yet show different symmetries. Particular to laser-induced plasma, algorithmic implementations of asymmetric Abel inversions exist [13–15], in which the emission intensity is decomposed into symmetric and asymmetric components. The method of asymmetric inclusion for this work is selected for the relative ease with which the method can be implemented. Consequences of applying the asymmetric Abel inversions are considered for application of inferring the plasma temperature from the measured, lateral AlO emission profiles.

## 2. Experiment

Spectra were collected following laser-induced breakdown initiated in ambient laboratory air using a Quantel Ultra Light Q-switched Nd:YAG laser with a pulse width of 8.5 ns, as measured by a fast silicon photodiode. The laser operated at the fundamental wavelength of 1064 nm with an average energy per pulse of 38 mJ at the ablation site. The ablation target was the narrow edge of 5 cm square aluminum alloy 6061 sheet that was approximately 1 mm thick. The AlO spectra are imaged onto the slit of a Princeton instruments Isoplane SCT 320 spectrometer installed with an 1800 grooves/mm grating and a focal length of 0.33 meters. Spatially and temporally resolved spectral images of the ablation event were recorded using a Princeton Instruments PiMax2 ICCD. The ICCD had a pixel arrangement of $256 \times 1024$ vertical by horizontal pixels. The horizontal axis was used to achieve the desired spectral resolution and the vertical axis was used to achieve spatial resolution along the plasma plume expansion height. Groups of 2 vertical pixels were binned together in order to improve the signal quality of the AlO emissions with respect to the measurement noise. The detector pixel size is 25 μm square such that in the binned configuration the effective pixel size was $50 \times 25$ μm. The imaging of the spectrometer-detector arrangement is such that the breakdown plasma is magnified at a ratio of 1:1.8. This was determined using a negative of the 1951 USAF target, a known and widely used target for image resolution testing consisting of images with known sizes. The spectral resolution of the spectrometer-detector arrangement was approximately 0.1 nm in the spectral region between 480 and 500 nm. The spectra used for analysis were the result of 100 accumulations on a single ablation site. The spectrometer-detector arrangement was calibrated for the system wavelength response using a Hg/Ne-Ar lamp and the system intensity response using a tungsten lamp with a known spectral intensity response curve. Time synchronization of the system was achieved by synchronizing the ICCD gate opening to the Q-switch of the laser.

## 3. Results and Discussion

Figure 1 shows contour mappings of spatially resolved measurements of AlO spectra between time delays of 20 and 35 µs following optical breakdown along the axis of laser incidence. The ablation surface is located at a slit height of 1.0 mm. Here we note that slit height units refer to the image plane measurement and not the exact positions of the spectral emissions in the plasma. Detailed ray tracings of the spectral imaging system (non trivial) would be required to correlate image plane units to the plasma emission source. Initially the AlO signal intensity is relatively isotropic between slit heights of 1.5 and 3.5 mm (Figure 1a). As the time delay increases, the spectroscopic intensity is seen to diminish, which is roughly correlated with the decay of the laser-induced plasma. As the time delay increases, areas of larger AlO signal intensity farther from the ablation surface, see 3.5–4 mm slit heights in Figure 1b. These areas of increased intensity also appear to be asymmetric, with stronger emissions occurring in the higher half of the plume, which is also the direction of the plasma plume and plasma formation blast wave propagation. The asymmetry of the axial, spectroscopic lines-of-sight likely indicates asymmetries in the radial profiles of the AlO emissions and the need for an asymmetric Abel deconvolution.

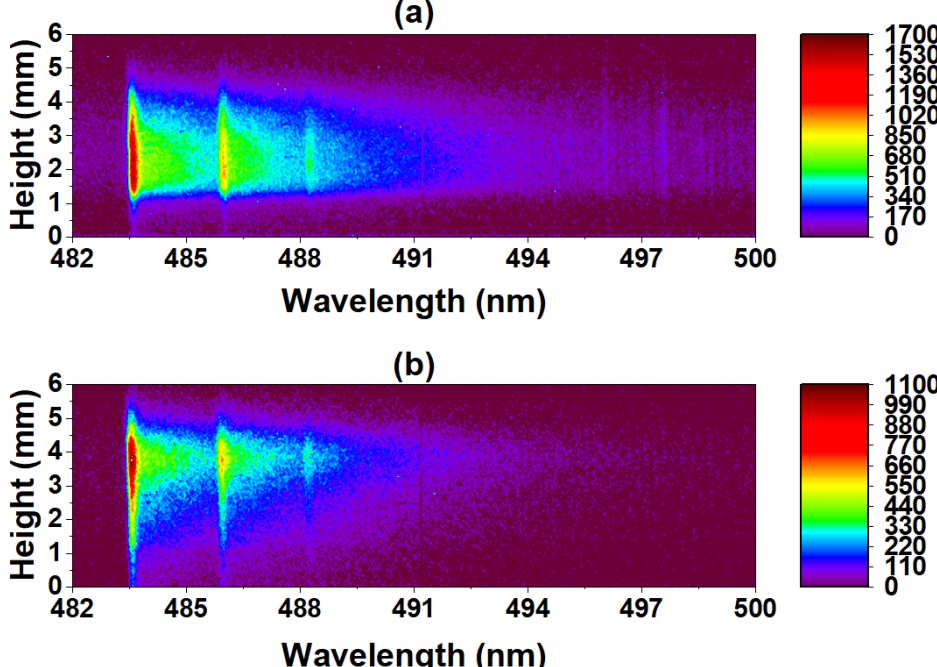

**Figure 1.** Contour mappings of spatially resolved AlO spectral measurements along the plasma plume height measured at (**a**) 20 µs and (**b**) 35 µs time delays.

The radial intensity distribution of plasma emissions are extracted through application of the Abel inversion under the assumption of cylindrical symmetry. Under such a scenario the Abel inverse transform is given by

$$I(r,\lambda) = \frac{-1}{\pi} \int_r^R \frac{dI(z,\lambda)}{dz} \frac{dz}{\sqrt{z^2 - r^2}}, \tag{2}$$

where $z$ is the lateral coordinate along the light of sight, $r$ is the radial coordinate, $I(z,\lambda)$ is the line-of-sight measurement, and $I(r,\lambda)$ is the deconvolved radial intensity profile.

In order to obtain an Abel inversion that accounts for minor asymmetries in the plasma spectral emissions, we consider the procedure introduced by Blades, in which Abel inversion symmetry is assigned to upper and lower plasma spectral regions [12]. These two regions are used to generate a symmetric intensity profile,

$$I_0(z, \lambda) = \frac{I(+z, \lambda) + I(-z, \lambda)}{2} \qquad (3)$$

by averaging the upper and lower plasma regions. The total inverted profile is given by

$$I(\pm r, \lambda) = G(\pm z, \lambda) I_0(r, \lambda) \qquad (4)$$

where $I_0(r, \lambda)$ is the Abel inversion of $I_0(z, \lambda)$ and $G(\pm z, \lambda)$ is given by

$$G(\pm z, \lambda) = \frac{I(\pm z, \lambda)}{I_0(r, \lambda)}. \qquad (5)$$

This factor accounts for the difference between the upper and lower half plasma regions and the symmetrized axial intensity distribution.

Application of the above procedure was carried out on the emission spectra depicted in the spectral contour mappings presented in Figure 1. The results of the asymmetric Abel inversion are presented in spatial-spectral contour maps in Figure 2. The method by which the Abel inversion of the symmetrized axial profile represented by Equation (2) uses a series expansion of orthogonal polynomials in which series expansion coefficients are determined through minimization techniques [16]. The orthogonal set of polynomials is taken to be Chebyshev polynomials of the first kind. The number of terms to be used in the procedure is 10. This maintains a decent fidelity of the deconvolved profile while avoiding potential oscillatory behavior and noise addition by considering significantly more series terms. The advantage of such a method is it alleviates the need to numerically compute the derivative presented in Equation (1) by way of the series expansion approximation. Such advantages exist in other Abel inversion implementations as well [17]. The Abel inversion is performed through use of a formalized MatLab script [18].

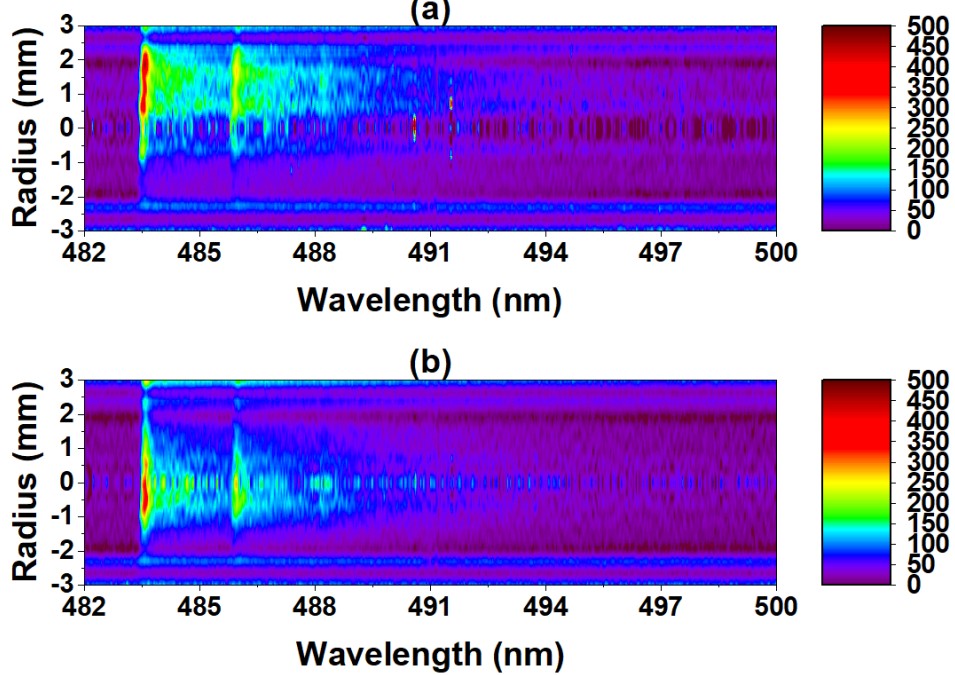

**Figure 2.** Contour mappings of extracted radial deconvolutions of measured AlO spectra at (**a**) 20 µs and (**b**) 35 µs time delays.

The largest source of error in the deconvolution procedure results from considering too small of a radius or applying the deconvolution to too small of an axial region. Attempting to apply the deconvolution procedure with a smaller axial region compared to the plasma radius will result in an inflated deconvolved signal intensity. By considering a smaller region of the plasma, oscillatory

behavior from the Abel inversion algorithm becomes prevalent in the radial profile. Also, edge effects tend to occur, whereby the radial deconvolution tends toward arbitrarily large values for the largest radius values in the deconvolution. Given the limitations in the size of an ICCD detector vs. the plasma image size, some of these effects may not be completely overcome in the outlined procedure. These effects were reduced in magnitude by applying the deconvolution procedure over most of the detector height, using slit heights from 0 to 6 mm. This was done since the AlO emission intensities are centered about a slit height of 3 mm. The calculated radial profiles correspond with this axial position.

The noisy nature of the deconvolution procedure is evident in the contour images of Figure 2, which is depicted by the fuzziness of the images. Potential edge effects are also apparent in each of the spectroscopic contour mappings and each favor a particular radial direction. Namely, the positive radial direction are favored. The positive direction corresponds to contributions from the lower plasma region. Reductions in quality of the deconvolution in this direction occur due to the presence of the aluminum surface.

Temperatures are determined from the molecular AlO emissions by fitting simulated AlO spectra to the line-of-sight measurements and radially deconvolved spectra. The theory spectra are simulated by making use of accurately compiled tables of line strengths for the diatomic molecule of interest [19]. In short, the intensity of a measured molecular line from upper state, $u$, to lower state, $l$, is given by

$$I_{ul} = \frac{16\pi^3 c (a_0 e)^2 C_{abs} N_0}{3\epsilon_0 Q} C_v \nu_u l^4 S_{ul} e^{hF_u/kT} \tag{6}$$

where $a_0$ is the Bohr radius, $e$, is the elementary charge, $c$ is the speed of light, $Q$ is the partition function, $N_0$ is the total population of the species, $C_{abs}$ and $C_v$ are the absolute and relative calibration factors, $h$ is Planck's constant, $F_u$ is the upper term value, $k$ is Boltzmann's constant, and $T$ is the temperature. The term $S_{ul}$ is the diatomic line strength and is calculated in a factorized form to account for electronic, vibrational, and rotational molecular structure. Detailed procedures for calculating a spectrum from the diatomic line strength are outlined in References [20–22].

Given the temperature dependence of the molecular spectrum, a fitting routine was used to extract the temperature of each temporally and spatially resolved AlO spectrum. Fitting was performed using a Trust-Region fitting routine with fitting parameters of temperature, line amplitude, and linear offset [23,24]. Additionally, the spectral resolution is a variable parameter that is used to establish the uncertainty of the inferred temperature from diatomic line profile fitting. Following initial fitting of each spectrum, the linear offset and spectral resolution parameters were randomly varied 1000 times and subsequently re-fit to generate a distribution of possible temperatures for each AlO spectrum. An additional fit was performed in which the line amplitude, temperature, linear offset, and spectral resolution were fit parameters. A total 1002 inferred temperatures were used to establish 1-$\sigma$ uncertainties according to the so-called Three Sigma Rule (68-95-99 Rule) for Gaussian distirbutions [25].

Figures 3 and 4 show calculated temperatures both in the lateral direction on the plasma plume height and in the deconvolved radial directions for 20 μs and 35 μs time delay following plasma initiation, respectively. These times were chosen to illustrate the different behaviors of the plasma expansions at two time delays that were as distinct as possible from each other. Furthermore, in order to maximize the number of spatial data points available for analysis, the later time delays were selected for analysis when the plasma had expanded to a sufficient spatial extent on the detector. Selecting later time delays also has the added benefit of avoiding hydrogen Balmer series beta line interferences which would severely impact and complicate molecular line profile fitting. It must also be stated that the two temporal points used for this study are insufficient for detailing the temporal decay of the temperature, though some manner of exponential decay is expected. The goal of this work is to investigate spatial temperature distributions. For the sake of validity of application of the asymmetric Abel inversion method to molecular emissions in laser ablation plasma, temperature inferences in which a symmetric Abel inversion was calculated are also presented. The shaded region surrounding

the symmetric Abel deconvolution (red dashed line) in Figures 3 and 4 represents the uncertainty of this temperature inference.

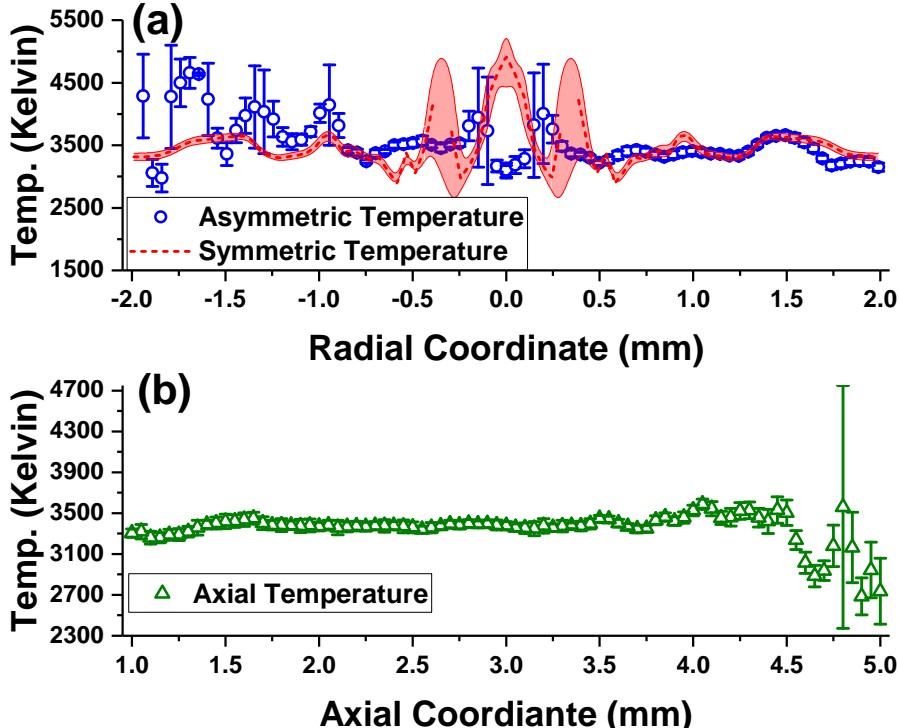

**Figure 3.** Temperatures inferred from AlO spectra at a time delay of 20 μs following ablation (**a**) along the plasma radius using asymmetric and symmetric Abel inverted spectra and (**b**) along the plasma using spatially resolved line-of-sight spectral measurements.

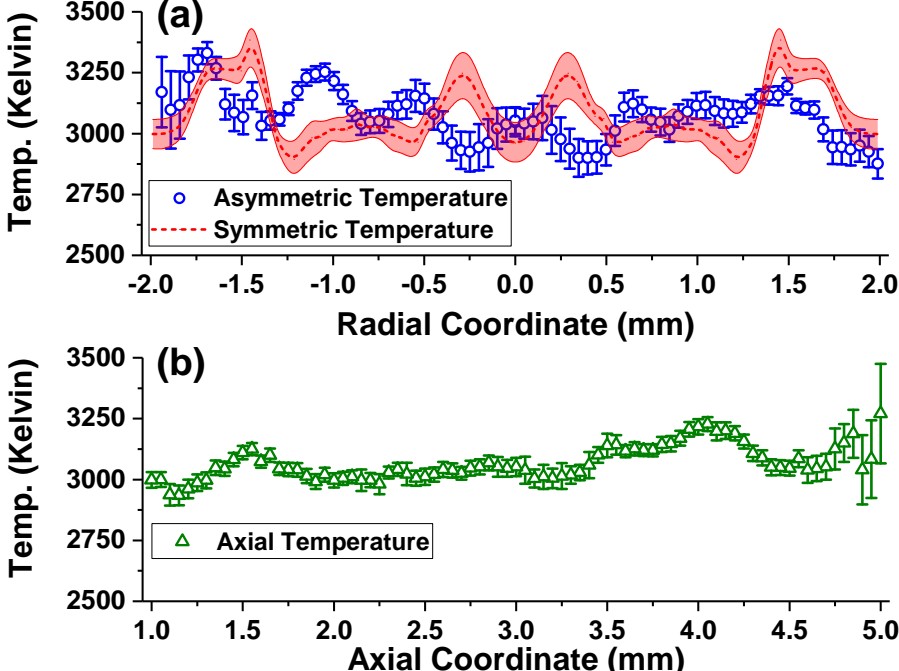

**Figure 4.** Temperatures inferred from AlO spectra at a time delay of 35 μs following ablation (**a**) along the plasma radius using asymmetric and symmetric Abel inverted spectra and (**b**) along the plasma using spatially resolved line-of-sight spectral measurements.

The axial temperatures at both 20, Figure 3b and 35, Figure 4b μs do not show much structure beyond the calculated 1-$\sigma$ uncertainties and show relatively constant temperatures of approximately 3400 and 3100 K, respectively, when averaging along the radial and axial directions. The displayed spectra and spatial temperature are expansions of central part of the plume. One must also note that the axial profiles represent the outward face of an expanding shell. In our case, the axial measurement shows consistency in the temperature profile. This is to be expected since the shock front is likely to have similar temperature and density conditions. The Abel inversion performed does not require a so-called "one to one map" for all features. Just because there is a particular type of variation in either the axial/radial profile does not mean there will be one of the exact shape in the corresponding radial/axial profile.

The spatial temperature distributions along the axial coordinates in Figures 3b and 4b indicate slightly higher temperatures near the boundaries, especially away from the target indicating the remnants of the laser-driven plasma excitation, but, again, one needs to consider this in view of the extent of the 1-$\sigma$ uncertainties. This is further discussed in the next paragraphs. Similarly, Figure 3a,b reveal slightly higher temperature near the edges, and with larger variation away from the target as indicated by the asymmetric results. The larger uncertainties in the negative radial directions correspond to weaker AlO signals in the upper most regions of the spatially resolved measurement (see Figure 1). The axial temperature expansion of the AlO molecules is of the order of 80–100 m/s, but the radial expansion appears to occur at lower speeds towards the plasma center.

Furthermore, the radial temperature profiles show more variation than the axial distributions. This could indicate that along the radial direction multiple shock fronts form behind the leading edge of the plasma expansion giving rise to more temperature profile structure when compared to the axial profile. The axial profile likely indicates a more even temperature distribution along a single front expanding along the line-of-sight. One should also consider that the Abel inversion process does add noise to the spectra, as has already been indicated. Though this noise is added directly to the spectra, it likely adds noise to the actual radial temperature distribution as well as evidenced by the increased uncertainties on the radial temperature inferences. Detailed discussion of the central discrepancies is expected to require additional data for analysis with Radon transforms.

For nominal 100 mJ, 10 ns pulsed laser radiation, the shock wave expands at speeds of the order of 1 mm/μs or 1000 m/s (Ma = 3) at 1 μs time delay. For 38 mJ pulses, the shock wave radius will be smaller according to the Taylor-Sedov blast wave model [26,27]. Moreover, for the investigated time delays of 20 and 35 μs, the Taylor-Sedov blast wave model predicts a shock wave radius of 2.7 and 3.4 mm, respectively, moving at speeds approaching the speed of sound (Ma = 1). Therefore, the axial temperature profiles show in part, the effects of the shock wave, indicated near 2.5 mm from center (or at the abscissa of 5 mm) in Figure 3b, but the shock wave appears to be just outside the investigated axial position in Figure 4b. Consequently, a temperature drop near the shock wave from about 3300 K to 2800 K (see Figure 3b) is quite reasonable. The radial profiles also indicate, in part, the shock wave effects near 2 mm. Moreover, it would not be unexpected to see an elevated temperature of AlO just inside the shock wave, with a slight indication of smaller temperatures at center.

## 4. Conclusions

In summary, we have shown that asymmetric Abel deconvolutional methods can be applied to obtain spatial information of molecular spectra in laser ablation plasma plumes. When coupled with diatomic line profile fitting of AlO emissions, we were able to demonstrate the impact that the propagating shockwave has on the plasma state in both radial and axial directions. Namely, multiple temperature gradients exist which correspond to the leading/following edges of the plasma blast wave. Further insight into the shockwave interaction with the plasma material may be obtained by coupling the radial deconvolution analysis with shockwave imaging studies, such as shadowgraph or schlieren techniques, or interferometric methods [28,29]. The purpose of such a study would be to

elucidate the affect of the plasma shockwave on the chemical kinetics of the plasma plume expansion into ambient atmospheres.

**Author Contributions:** All authors contributed equally to this work.

**Funding:** This paper describes objective technical results and analysis. Any subjective views or opinions that might be expressed in the paper do not necessarily represent the views of the U.S. Department of Energy or the United States Government. Sandia National Laboratories is a multimission laboratory managed and operated by National Technology and Engineering Solutions of Sandia, LLC., a wholly owned subsidiary of Honeywell International, Inc., for the U.S. Department of Energy's National Nuclear Security Administration under contract DE-NA0003525.

**Conflicts of Interest:** The authors declare no conflict of interest.

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
