# Peer review of "Spatial Molecular AlO Temperature Distributions in Laser-Induced Plasma"

_atoms, doi:10.3390/atoms7030086_

Round 1

Author Response

Reponses to Reviewer 1:

All responses to Reviewer 1 in the revised manuscript are indicated with text in color green. Where necessary, line numbers for the revised manuscript are included. Details are provided below:

Page 1, in abstract line 7, remove “we measured” and add were measured after “plasma initiation” in line 8 before the period.

The word has been replaced.

Figures 3 and 4; points/error bars are cut off toward the end of the x-axis:

1) 3a-last point shows half of a circle rather than full circle

2) 3b—points between 4.5 and 5 mm on the axis have error bars missing either tops or bottoms, y-axis need to be expanded to them better; also last point, show half of a triangle and error caps.

3) 4a-last point show a half of a circle and error caps

4) 4b-last point show a half of a triangle and error caps

5) The axes on these graphs need to be fixed to show the full points and errors bars

Figures 3 and 4 have been edited such that all points and error bars are not cut off. The figure filenames have been renamed to <Fig3rev.pdf> and <Fig4rev.pdf>.

Why were the time delays of 20 and 35 ms chosen? This is well beyond where the LIBS emission is best. Why not use a 1 ms time delay, where the emission is much stronger and less noisy? I would think the overall calculated temperatures would be lower since the data was taken when the plasma is cooling down. Would this model be able to predict the temperatures using the time delays where the emission is best?

The term “emission is best” is highly subjective. Commonly, researchers will try to “optimize” a LIBS measurement by selecting an observation delay that provides the best signal for a particular spectral feature (eg. C 247, Zn 481, Halpha, et c.). However, in many instances, LIBS experiments are performed at a particular time after plasma initiation but with large integration/exposure times (~100’s of microseconds to milliseconds) in order collect an optimal signal. Such a typical experimental methodology is particularly necessary for instruments that lack an intensifier on the detector chip in enhance low light signals from short gate time measurements. During the extended time of a long exposure time, the dynamics of the plasma expansion are occurring in addition to the analytical value of the spectral lines.

The selected time delays reflect the need to consider that the plasma has to appropriately cool for molecular recombination to occur, which depending on plasma initiation conditions and the particular molecule can occur anywhere between 500 ns to several microseconds following plasma initiation. We also needed to consider that the best spatial deconvolution would occur (more data points) when our detector measured the largest spatial distribution which occurs when the plasma has decayed for some time. 20 and 35 microseconds were selected as two distinct time delays that meat this condition. The late time delays also had the added feature of avoiding hydrogen Balmer beta line interferences. This sentiment is expressed on lines 154-158.

In regards to predicting temporal temperature decay, two times are insufficient for this purpose, though the decaying temperature trend has been measured previously (see refs. 2, 21, and 22 for example). The point of this study is study the spatial distribution of the temperatures inferred from molecular line profile fitting. We have highlighted the direct scope of our work on lines 158-161.       

 Figure 3 and 4 explanations need to include discussions on the following:

Why do the spatially resolved line-of-sight figures (3b and 4b) show less fluctuations in the temperatures (between 1 mm and 4.5 mm) over the Abel inversion figures (3a and 4a)?

The axial profiles represent the outward face of an expanding shell. In our case, the axial measurement shows consistency in the temperature profile. This is to be expected since the shock front is likely to have similar temperature and density conditions. The Abel inversion performed does not require a so-called “one to one map” for all features. Just because there is a particular type of variation in either the axial/radial profile does not mean there will be one of the exact shape in the corresponding radial/axial profile. We have added text to the revised manuscript on lines 162-168 to reflect this sentiment.

There is an outlier point in all figures located at about -1.75 mm in the (a) figures and about 1.25 mm in the (b) figures. What happened at that point?

I do not believe there is an outlier at this point, rather this point which appears at the same place in each Figs. 3a/b and 4a/b is part of the legend. The x-axis positions provided reviewer 1 correspond to these points. To help readers avoid this potential confusion a box has placed around the legends of Fig 3a/b and Fig 4a/b in the revised figures.

3b and 4b , they are relatively constant as the discussion states on page 5, lines 158-160 There is significant variation in figure 3b between 4.5 and 5 mm and in 4b between about 4.75 and 5 mm. This is explained in the next paragraph with the shock wave analysis. Add a statement that saying that the temperature variations seen between 4.5 and 5 mm and 4.75 and 5 mm in Figures 3b and 4b, respectively, are discussed in the next paragraph or later.

Added: This is further discussed in the next paragraphs. (new line 179)

Figure 3a:

It is stated there is slightly higher temperatures near the boundaries especially away from the targets. This doesn’t seem completely accurate if you take in account the error bars. The edge near -2 mm and points at -0.25 and +0.25 mm show very similar asymmetric temperatures and at the edge at 2 mm the asymmetric temperature is lower and on par with the 0 mm position.

Though we already state that the error bars need to be considered when interpreting the spatial axial temperature distribution on lines 166-167, we reiterate this point again on line 178 and 179.

There seems to more fluctuations between -2 and 0.25 mm for the asymmetric data with it leveling off between 0.25 and 2 mm.

We agree, the data indicate this trend.

The symmetric data shows significantly higher temperatures near the center positions between (-0.5 and +0.5): Why is that the case when compared to the asymmetric data in Figure 3 which predict lower temperatures in that range? Also, why is that the case here when compared to what the symmetric data for 4a where the trend looks opposite?

For Figure 3, the 1-sigma error bars tend to indicate overlap of the symmetric and asymmetric analysis. In turn, for Figure 4, symmetric and asymmetric results tend to agree near the boundaries. Detailed discussion of the central discrepancies is expected to require additional data for analysis with Radon transforms. The last sentence has been added in the manuscript just above the last paragraph before Section 4.

Figure 4a :

4a does show higher asymmetric temperatures at -2 mm and significantly lower asymmetric temperature at 2 mm; state that this is discussed in the next paragraph or later too.

We previously added: This is further discussed in the next paragraphs. (new line 179) - and we feel that the previous addition would imply the same thing, so we avoided repetition.

There seems to be a lot more fluctuation in both the asymmetric and symmetric temperature data using the radial coordinates. Why is that?

This is a result of the Abel inversion and likely indicates that radial structure fluctuates more than axial temperature. The difference between the two is the axial temperature distribution is an indication of the temperature along a plane parallel to the shock front and the radial the temperature distribution gives details a plan perpendicular to the shock front. More variation in this plane likely indicates the presence an initial shock front and subsequent shock fronts behind the leading edge of the expansion, though this needs to be viewed in consideration of the fact that the Abel inversion adds noise to the AlO spectra and possibly the radial temperature distribution. This sentiment is added to the manuscript on lines 185-192.

A discussion of the shock wave differences from 100 mJ using a 1 delay is discussed in last paragraph (pg 6 line 170-171):  A statement is made about 38 mJ pulses and how its shock wave is smaller. Why discuss the 100 mJ data using 1 ms time delay at all? It does pertain to the paper’s results. This discussion should explain the general differences of how the shock waves are affected by both the time delay and energy and then it should go into discussing the 38 mJ data with the two time delays chosen.

We reference the Taylor-Sedov model and feel that the results are accordingly debated to be in agreement with the parametric dependence on energy/pulse and time delay.

Reviewer 2 Report

Line 68: ‘The’ should be ‘This’, in fact this sentence needs to be clearer.

line 131: Complied should be compiled

line 133: Electronic should be elementary 

line 161: missing ‘are’ in front of ‘expansions’

Line 187: missing ‘be’ after ‘may’ 

Author Response

All responses to Reviewer 2 in the revised manuscript are indicated with text in color blue. Details are provided below:

Line 68: ‘The’ should be ‘This’, in fact this sentence needs to be clearer.

The grammar mistake has been fixed and the sentenced has been clarified to indicate that 1951 USAF target is a standard imaging target used to resolution testing consisting of images of known sizes.

line 131: Complied should be compiled

The spelling error has been corrected.

line 133: Electronic should be elementary 

The word has been changed.

line 161: missing ‘are’ in front of ‘expansions’

This missing word has been inserted.     

Line 187: missing ‘be’ after ‘may

The missing word has been inserted.